# Incidence of pelvic fractures after definitive radiotherapy for cervical cancer: A retrospective multicenter cohort study (The IPFAR study)

Shinsuke Shirakawa[1], Shoji Nagao[2]*, Kotaro Yoshio[3], Toshiharu Mitsuhashi[4], Noriaki Imafuku[5], Dan Yamamoto[5], Masae Yorimitsu[6], Junichi Kodama[6], Hisako Nagasaka[7], Yoshie Nakanishi[7], Hisashi Masuyama[1]

**1** Department of Obstetrics and Gynecology, Okayama University, Kita-ku, Okayama, Japan,
**2** Department of Obstetrics and Gynecology, Faculty of Medicine, Dentistry and Pharmaceutical Sciences, Okayama University, Kita-ku, Okayama, Japan, **3** Department of Radiology, Okayama University, Kita-ku, Okayama, Japan, **4** Center for Innovative Clinical Medicine, Okayama University, Kita-ku, Okayama, Japan, **5** Department of Obstetrics and Gynecology, Fukuyama Medical Center, Fukuyama, Hiroshima, Japan, **6** Department of Obstetrics and Gynecology, Hiroshima City Hiroshima Citizens Hospital, Naka-ku, Hiroshima, Japan, **7** Department of Obstetrics and Gynecology, Kagawa Prefectural Central Hospital, Takamatsu, Kagawa, Japan

* s_nagao@okayama-u.ac.jp

## Abstract

### Objective

This multicenter study aimed to identify the risk factors for pelvic insufficiency fractures (PIFs) in women who received definitive radiotherapy (RT) as the initial treatment for cervical cancer and to examine the differences in the incidence of PIFs across institutions.

### Patients and Methods

The medical records of 208 women were reviewed. These women received definitive RT as an initial treatment for cervical cancer at four institutions between January 2016 and December 2018.

### Results

The median age was 61.5 years (range: 29–93 years). Overall, 59 patients (28.4%) developed PIF, with 48 (81.4%) of them developing it within two years after completion of RT. Multivariate analysis identified postmenopausal status and treating institution as independent risk factors. The incidence of PIF varied significantly among institutions, ranging from 18.9% to 50.0%, despite no significant differences in patient demographics or RT protocols.

**Data availability statement:** All relevant data are within the paper and its Supporting Information files.

**Funding:** The author(s) received no specific funding for this work.

**Competing interests:** The authors have declared that no competing interests exist.

## Conclusion

Substantial inter-institutional variation in PIF incidence was observed, even under standardized treatment conditions. These findings underscore the need for individualized risk assessment and institutional quality control in the long-term management of cervical cancer survivors.

## Introduction

Radiotherapy (RT) for cervical cancer causes pelvic insufficiency fractures (PIFs) in 10%–29% of patients [1–5]. More than 80% of these fractures develop within two years of RT completion, with symptoms such as chronic pain occurring in the majority of cases [6]. Chronic pain not only compromises the long-term quality of life (QOL) of cancer survivors but also frequently necessitates the use of opioid analgesics for adequate pain management [6]. Several studies have identified advanced age, postmenopausal status, and low body mass index (BMI) as significant risk factors for the development of PIFs [6–9]. Furthermore, definitive RT itself has been recognized as a major contributing factor, although this risk may be reduced with the increasing adoption of advanced techniques such as volumetric modulated arc therapy (VMAT) and intensity-modulated radiation therapy (IMRT) [6,9].

Although numerous studies have investigated PIFs following RT for cervical cancer, the reported incidence rates vary widely [1–5]. Most of these studies have been conducted at single institutions, and the observed heterogeneity is likely attributable to differences in patient demographics, racial and regional characteristics, medical histories, RT protocols, and diagnostic criteria for PIFs. Such variability hampers direct inter-study comparisons and may mask underlying epidemiological trends.

To address these limitations, we conducted a multicenter retrospective cohort study to examine inter-institutional variability in the incidence of PIFs under controlled conditions. The four participating institutions are located in geographically adjacent areas of Japan and follow comparable RT protocols. To ensure diagnostic consistency, all PIF cases were re-evaluated based on standardized radiologic criteria. The primary aim of this study was to determine whether differences in PIF incidence persist among institutions even after adjusting for patient- and treatment-related variables. If such variability remains, we further sought to explore potential contributing factors. By focusing on inter-institutional differences under standardized conditions, this study aims to provide new insights into the epidemiology of PIFs following RT for cervical cancer, extending beyond the scope of previous single-institutional reports.

## Patients and methods

This study was a multicenter cohort study that included patients with cervical cancer who received RT.

### 1. Patients

Four institutions participated in this multicenter collaboration: Okayama University Hospital, Hiroshima City Hospital, Fukuyama Medical Center, and Kagawa

Prefectural Central Hospital. All participating institutions are located within a 200-kilometer radius in the Setouchi region of Japan and maintain close collaboration in both clinical practice and personnel exchange.

Eligible patients were those newly diagnosed with pathologically confirmed cervical cancer between January 2016 and December 2018 at the participating institutions. All patients received definitive RT, with or without concurrent chemotherapy (CCRT), as initial treatment. Patients who underwent nodal boost, high-dose-rate intracavitary brachytherapy (HDR-ICBT), central shielding (CS), VMAT, or IMRT were eligible for inclusion in the analysis. Exclusion criteria included patients who received postoperative adjuvant RT, those treated with RT for recurrent disease, patients who did not complete the scheduled RT, and patients who were not appropriately followed up after treatment.

## 2. Radiotherapy protocols across participating institutions

Table 1 provides an overview of the standardized radiotherapy protocols implemented across the participating institutions. All centers conformed to the Japanese national guidelines for radiotherapy, and no notable inter-institutional differences were observed with regard to radiotherapy equipment [10]. In all patients, treatment consisted of a combination of external beam radiation therapy (EBRT) and HDR-ICBT, delivered using 10 MV photon beams.

The clinical target volume (CTV) for EBRT encompassed the primary cervical tumor, entire uterus, bilateral parametria, the upper half of the vagina, and regional pelvic lymph nodes, including the common, internal, and external iliac, obturator, and presacral nodes. All institutions employed whole-pelvis (WP) irradiation using a conventional four-field box technique, followed by anterior–posterior parallel-opposed fields with CS. The cumulative pelvic dose (WP plus CS) ranged from 46.8 to 54.0 Gy. HDR-ICBT was administered in four fractions, with a total dose of up to 24 Gy.

While EBRT protocols were largely uniform across institutions, variation was noted in the management of para-aortic lymph node (PAN) irradiation. Although the four-field box technique was generally used, Institutions C and D uniquely employed VMAT for PAN coverage.

**Table 1. Radiotherapy protocols across participating institutions.**

| | Institution A | Institution B | Institution C | Institution D |
|---|---|---|---|---|
| Upper edge of WP irradiation field | Aortic bifurcation (L4-5 level) | Aortic bifurcation (L4-5 level) | Aortic bifurcation (L4-5 level) | Aortic bifurcation (L4-5 level) |
| Upper edge of PAN irradiation field | Upper edge of left renal vein | Upper edge of L1 | Upper edge of L1 | Upper edge of left renal vein |
| Lower edge of PAN irradiation field | Aortic bifurcation | Aortic bifurcation | Aortic bifurcation | Aortic bifurcation |
| Left and right edges of PAN irradiation field | 10-20 mm expansion from aorta/IVC laterally | About 10 mm expansion from aorta/IVC laterally | 10mm expansion from aorta/IVC laterally | 10-20 mm expansion from aorta/IVC laterally |
| Ventral edge of PAN irradiation field | 10-20 mm expansion from aorta/IVC laterally | About 10 mm expansion from aorta/IVC forward | 10mm expansion from aorta/IVC laterally | 10-20 mm expansion from aorta/IVC laterally |
| Dorsal edge of PAN irradiation field | 10-20 mm expansion from aorta/IVC laterally | About 10 mm expansion from aorta/IVC dorsal side | 10mm expansion from aorta/IVC laterally | 10-20 mm expansion from aorta/IVC laterally |
| WP irradiation field method | 4-field box technique | 4-field box technique | 4-field box technique | 4-field box technique |
| PAN irradiation field method | 4-field box technique | 4-field box technique | VMAT | 4-field box technique or VMAT |
| The position of CS | S2/3 level to cover the presacral area. | S2/3 level to cover the presacral area. | S2/3 level to cover the presacral area. | S2/3 level to cover the presacral area. |
| Type of instrument | Primus; Canon Medical Systems, Tochigi, Japan | CLINAC iX; Varian Medical Systems, USA | True Beam; Varian Medical Systems, USA | Synergy; Elekta Medical Systems, Stockholm. Sweden |

WP, whole pelvis; PAN, para-aortic lymph nodes; IVC, inferior vena cava; VMAT, volumetric modulated arc therapy; IMRT, intensity modulated radiation therapy; CS, center shield.

## 3. Follow-up

After the completion of RT, according to the clinical guideline, the all patients underwent medical examinations approximately every three months for the first two years and then every six months for the following three years. Radiological imaging tests, including computed tomography (CT) and magnetic resonance imaging (MRI), were performed every six months for the first two years and then every 12 months for the following three years. Radiological imaging was performed routinely during follow-up regardless of symptoms thought to be related to PIF., and both symptomatic and asymptomatic PIFs detected on surveillance imaging were included in the analysis. PIF was defined as a fracture occurring within the irradiated pelvic field without evidence of tumor recurrence.

## 4. Data collection

Relevant clinical information was extracted from medical records. This included patient characteristics, disease details, treatment regimens, and the occurrence of PIFs. For this study, all CT and MRI images were initially reviewed by a radiologist at each institution. Subsequently, these images were double-checked by a gynecologic oncologist at Institution A, who was trained by a radiologist, based on standardized diagnostic criteria established by that radiologist. PIF was diagnosed when fracture lines were observed on CT or MRI within the irradiated field, as described above. Both symptomatic and asymptomatic PIFs detected on surveillance imaging were included in the analysis. PIF was defined as a fracture occurring within the irradiated pelvic field without evidence of tumor recurrence. All confirmed PIF cases demonstrated fracture lines on CT or MRI within the radiation field. Some fractures appeared as subtle lines accompanied by slight changes in bone signal intensity (referred to as "coloration"). However, these were classified as PIF only when corresponding structural abnormalities consistent with a fracture were clearly present, even if minimal. Lesions lacking such findings or those suggestive of metastasis were excluded from the analysis.

## 5. Data analysis

The collected data were analyzed under the supervision of a biostatistician. Descriptive statistics were calculated for both PIF and non-PIF groups. The Mann-Whitney U test was used for comparison. We performed further descriptive statistics for the number of patients, age, menopausal status, BMI, medical history, drug history, FIGO (International Federation of Gynecology and Obstetrics) stage (2018), combination therapy with chemotherapy, PIF rate, and painkiller use at each institution. We calculated the PIF rate after RT at each institution and analyzed the data using the Kaplan–Meier method. PIF was treated as an event, and cases of death or disease progression were censored. Cox proportional hazards regression modeling was used to evaluate the association of clinical factors with PIF in multivariate analysis. To adjust for potential bias, inverse probability weighting was applied. Statistical analyses were performed using IBM SPSS Statistics (version 29.0; IBM Corp., Armonk, NY). Statistical significance was set at $p < 0.05$.

## 6. Ethical Considerations

This study was conducted in accordance with the Declaration of Helsinki. All participants were assured that their personal identities would remain confidential and that participation was voluntary, with the option to opt out at any time. The study was approved by the Clinical Research Ethics Review Committee of Okayama University Hospital (Approval No.: 2203−045).

For research purposes, the medical records and imaging data were accessed between May 1, 2022, and December 31, 2022. During data collection, the investigators had access to potentially identifiable information (such as hospital ID numbers and clinical records). However, all data were anonymized before analysis, and no personally identifiable information (e.g., names or addresses) was included in the dataset used for statistical analyses.

## Results

### 1. Incidence and site of PIF

A total of 208 patients who underwent definitive RT as the initial treatment for cervical cancer between January 2016 and December 2018 were enrolled in this study (81, 56, 37, and 34 patients in Institutions A, B, C, and D, respectively). The median follow-up duration was 36 months (range: 1–72 months). Overall, 28.4% (59/208) of the patients developed PIF in the irradiated field.

Fracture sites among the 59 affected patients were distributed as follows: pelvis in 34 cases (57.6%), lumbar spine in 13 (22.0%), femoral head or neck in 4 (6.8%), and unspecified sites in 8 (13.6%). Of the patients who developed PIFs, 31 (52.5%) reported chronic pain necessitating long-term use of analgesics. Representative radiological images of PIFs, including anatomical fracture sites and corresponding radiation dose distribution maps, are presented in Fig 1. These images demonstrate a close spatial association between the location of fractures and the high-dose irradiation fields, suggesting a dose-dependent relationship in the development of PIFs.

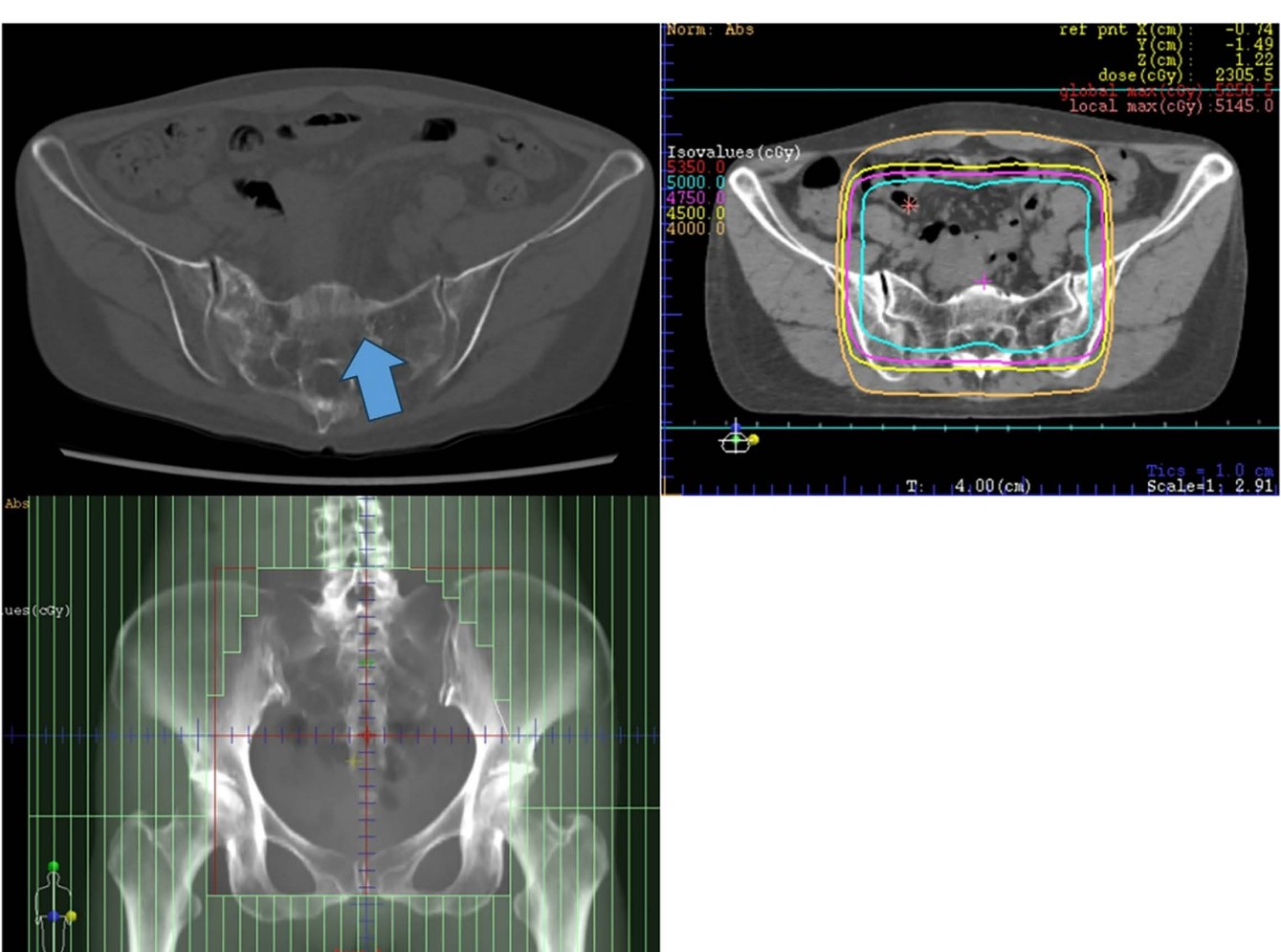

**Fig 1. Representative CT of PIF and corresponding RT dose distribution.** The arrow indicates the site of a PIF in the sacrum, as seen on CT. The RT dose distribution for the same patient is also shown, demonstrating that the fracture occurred within the high-dose irradiated field.

## 2. Onset of PIF

Among the 59 patients who developed PIF, 48 (81.4%) were diagnosed within two years after completion of RT (Fig 2). PIF was diagnosed at a median of 12 months (range, 1–51 months) after RT. In institutions A and C, almost all patients developed PIF within two years of completing RT, with no significant difference. Conversely, in institutions B and D, patients developed PIF within 4 years of completing RT (Fig 3).

## 3. Baseline characteristics of the study cohort

The characteristics of women with and without PIF are presented in Table 2. The median age of the entire cohort was 61.5 years, and 69.7% of the patients were postmenopausal. Only a small proportion of patients were receiving oral medications at the initiation of radiotherapy, including hormone replacement therapy (e.g., estradiol) or anti-osteoporotic agents. Significant differences in age, menopausal status, and institutional affiliation were observed between the PIF and non-PIF patients. The number of patients who developed PIF was 19.8% (16/81), 33.9% (19/56), 18.9% (7/37), and 50% (17/34) at institutions A, B, C, D, respectively, with significant differences among institutions (p = 0.004).

Table 3 presents the characteristics of these institutions. Institutions B and C had a relatively higher proportion of patients with advanced FIGO stage disease. In contrast, no significant inter-institutional differences were observed regarding menopausal status or BMI, both of which are recognized risk factors for PIF.

Table 4 presents the treatment characteristics of each institution. There was a certain degree of inter-institutional difference in irradiation method (P = 0.01), with PAN irradiation being more frequently performed at Institution D, and whole

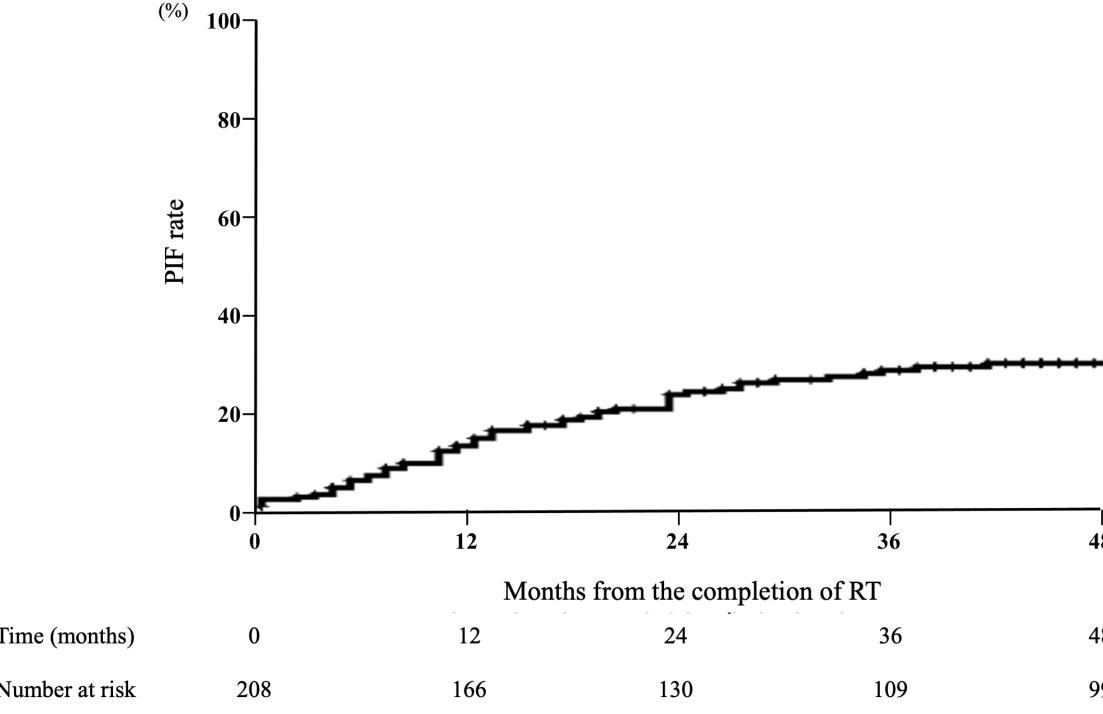

| Time (months) | 0 | 12 | 24 | 36 | 48 |
|---|---|---|---|---|---|
| Number at risk | 208 | 166 | 130 | 109 | 99 |

**Fig 2. Incidence of PIF after the end of RT for cervical cancer (N = 208).** The timing of PIF onset was analyzed in 59 patients who developed PIF. The x-axis represents the number of months following the completion of radiation therapy, and the y-axis shows the cumulative percentage of patients with PIF. A total of 81.4% of PIFs occurred within two years after completion of RT.

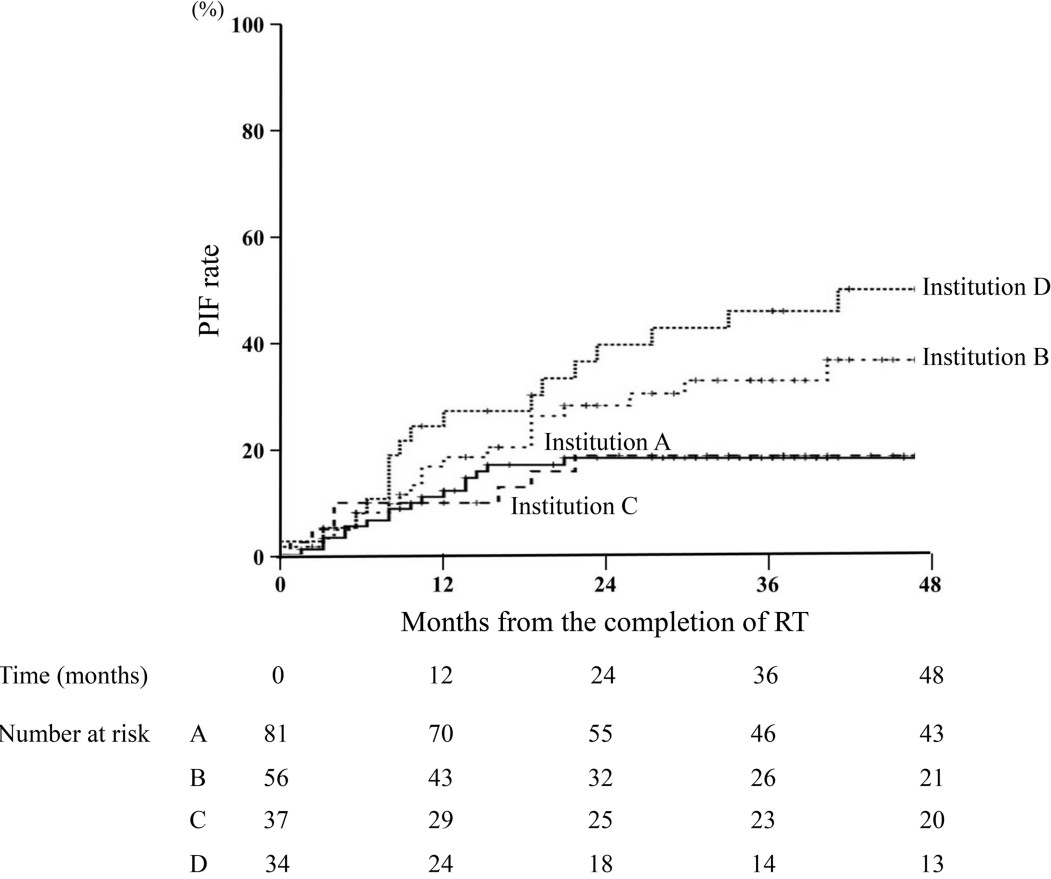

| Time (months) | | 0 | 12 | 24 | 36 | 48 |
|---|---|---|---|---|---|---|
| Number at risk | A | 81 | 70 | 55 | 46 | 43 |
| | B | 56 | 43 | 32 | 26 | 21 |
| | C | 37 | 29 | 25 | 23 | 20 |
| | D | 34 | 24 | 18 | 14 | 13 |

**Fig 3. Incidence of pelvic insufficiency fracture in each institution.** Among patients who developed pelvic insufficiency fracture (PIF), this figure shows when the PIF occurred in patients in each institution. The horizontal axis represents the month PIF occurred, and the vertical axis represents the percentage of PIFs. In Institution A and C, almost all patients developed PIF within 2 years after completion of RT, with no significant difference. On the other hand, in Institution B and D, patients developed PIF within 4 years after RT. **(A)** Institution A (N = 81) **(B)** Institution B (N = 56) **(C)** Institution C (N = 37) **(D)** Institution D (N = 34).

pelvic RT + HDR-ICBT being more frequently performed at Institution A. Furthermore, although there was no significant difference, Institution A tended to perform boost irradiation to the lymph nodes more frequently.

### 4. Multivariate analysis of risk factors for PIF

The following variables were included in the multivariate model based on clinical relevance and presumed independence: menopausal status, BMI, histological subtype, use of CCRT, PAN irradiation, HDR-ICBT, and treating institution. Comorbidities such as rheumatoid arthritis, osteoporosis, breast cancer, prior fractures, and use of hormone replacement or anti-osteoporotic medications were excluded due to their low prevalence. Although BMI < 23 kg/m² was not statistically significant in univariate analysis (p = 0.081), it was included based on previous reports identifying low BMI as a potential risk factor for PIF. In this analysis, however, BMI was not an independent predictor. Among the variables assessed, only menopausal status and treating institution were significantly associated with PIF (Table 5).

The hazard ratio (HR) for menopause was 7.391 (95% CI: 2.63–20.79), and for Institution D (vs. Institution A), the HR was 1.214 (95% CI: 1.015–1.412). Comparing high-incidence institutions (B and D) with low-incidence ones (A and C), the composite HR for PIF was 1.870 (95% CI: 1.069–3.269). Institution D, with the highest PIF incidence, also had a greater

**Table 2. Baseline characteristics of patients with and without PIF.**

| | PIF (N = 59) | Non-PIF (N = 149) | p-value |
|---|---|---|---|
| Median age (range) | 71 (42–90) | 54 (29–93) | <0.001 |
| Postmenopausal state | 55 (93.2%) | 90 (60.4%) | <0.001 |
| BMI < 23 kg/m² | 16 (27.1%) | 60 (40.3%) | 0.081 |
| Medical history | | | |
| Rheumatoid arthritis | 1 (1.7%) | 2 (1.4%) | |
| Osteoporosis* | 0 | 3 (2.1%) | |
| Breast cancer | 0 | 1 (0.7%) | |
| Insufficiency fracture | 4 (6.8%) | 7 (4.7%) | 0.953 |
| FIGO stage | | | |
| I | 8 (13.6%) | 21 (14.1%) | |
| II | 14 (23.7%) | 57 (38.3%) | |
| III | 28 (47.4%) | 50 (33.6%) | |
| IV | 9 (15.3%) | 21 (14.1%) | 0.192 |
| Histopathology | | | |
| Squamous cell carcinoma | 45 (76.3%) | 126 (84.6%) | |
| Adenocarcinoma | 13 (22.0%) | 18 (12.1%) | |
| Adenosquamous carcinoma | 0 | 2 (1.4%) | |
| Unknown | 1 (1.7%) | 3 (2.1%) | |
| Combination chemotherapy | 41 (69.5%) | 118 (79.2%) | 0.275 |
| Irradiation range, Using HDR-ICBT | | | |
| WP | 5 (8.5%) | 15 (10.1%) | |
| WP + PAN | 4 (6.8%) | 2 (1.4%) | |
| WP + HDR-ICBT | 39 (66.1%) | 112 (75.2%) | |
| WP + PAN + HDR-ICBT | 11 (18.6%) | 20 (13.4%) | 0.124 |
| Institution | | | |
| A | 16 (27.1%) | 65 (43.6%) | |
| B | 19 (32.2%) | 37 (24.8%) | |
| C | 7 (11.9%) | 30 (20.1%) | |
| D | 17 (28.8%) | 17 (11.4%) | 0.004 |

PIF, pelvic insufficiency fracture; WP, whole pelvis; PAN, para-aortic lymph nodes; HDR-ICBT, high-dose-rate intracavitary brachytherapy. Mann-Whitney U test was applied for comparison.

*All three patients had a history of bisphosphonate therapy.

proportion of patients receiving PAN irradiation. No consistent inter-institutional differences were found in the use of central shielding or nodal boost.

## Discussion

In this multicenter cohort study, we investigated 208 cervical cancer patients treated with definitive RT and found a relatively high overall incidence of PIFs at 28.4%. The incidence varied substantially across institutions, ranging from 18.9% to 50.0%. Notably, over 80% of PIFs occurred within two years post-RT, and more than half of affected patients experienced chronic pain requiring long-term analgesic use. Multivariate analysis identified postmenopausal status and treating institution as independent predictors of PIF development. Other conventional risk factors, such as low BMI and histological subtype, were not statistically significant.

**Table 3. Baseline characteristics of patients of each instituion.**

| | Institution A (N=81) | Institution B (N=56) | Institution C (N=37) | Institution D (N=34) | p-value |
|---|---|---|---|---|---|
| Median age (range) | 58 (35–91) | 68.5 (30–93) | 59 (29–80) | 63 (35–86) | 0.265 |
| Postmenopausal state | 54 (66.7%) | 40 (71.4%) | 27 (73.0%) | 24 (70.6%) | 0.889 |
| BMI<23 kg/m$^2$ | 30 (37.0%) | 20 (35.7%) | 15 (40.5%) | 11 (32.4%) | 0.907 |
| Medical history | | | | | |
| Rheumatoid arthritis | 2 (2.5%) | 0 | 1 (2.7%) | 0 | |
| Osteoporosis | 0 | 1 (1.8%) | 0 | 2 (5.9%) | |
| Breast cancer | 1 (1.2%) | 0 | 0 | 0 | |
| PIF | 2 (2.5%) | 5 (8.9%) | 1 (2.7%) | 3 (8.8%) | |
| FIGO stage | | | | | |
| I | 17 (21.0%) | 3 (5.4%) | 5 (13.5%) | 4 (11.8%) | |
| II | 38 (46.9%) | 15 (26.8%) | 6 (16.2%) | 12 (35.3%) | |
| III | 22 (27.1%) | 24 (42.9%) | 19 (51.4%) | 13 (38.2%) | |
| IV | 4 (4.9%) | 14 (25.0%) | 7 (18.9%) | 5 (14.7%) | 0.001 |
| Histopathology | | | | | |
| SCC | 64 (79.0%) | 44 (78.6%) | 33 (89.2%) | 30 (88.2) | |
| Adenocarcinoma | 14 (17.3%) | 9 (16.1%) | 4 (10.8%) | 4 (11.8) | |
| Adenosquamous | 1 (1.2%) | 1 (1.8%) | 0 | 0 | |
| Unknown | 2 (2.5%) | 2 (3.6%) | 0 | 0 | 0.737 |
| CCRT | 61 (75.3%) | 39 (69.6%) | 32 (86.5%) | 27 (73.0%) | 0.5 |

PIF, pelvic insufficiency fracture; SCC, squamous cell carcinoma; CCRT, concurrent chemoradiation therapy. Continuous variables were compared using the Mann–Whitney U test and categorical variables using the chi-square test. Values are descriptive unless otherwise indicated.

**Table 4. Treatment characteristics of each institution.**

| | Institution A (N=81) | Institution B (N=56) | Institution C (N=37) | Institution D (N=34) | p-value |
|---|---|---|---|---|---|
| Irradiation range, Using | | | | | |
| HDR-ICBT | | | | | |
| WP | 4 (4.9%) | 6 (10.7%) | 8 (21.6%) | 2 (5.9%) | |
| WP+PAN | 0 | 2 (3.6%) | 2 (4.4%) | 2 (5.9%) | |
| WP+HDR-ICBT | 67 (82.7%) | 41 (73.2%) | 25 (67.6%) | 18 (52.9%) | |
| WP+PAN+HDR-ICBT | 10 (12.3%) | 7 (12.5%) | 2 (5.4%) | 12 (35.3%) | 0.001 |
| WP/CS (Gy) | | | | | |
| 30/20 | 70 (86.4%) | 6 (10.7%) | 2 (5.4%) | 9 (26.5%) | |
| 40/10 | 6 (7.4%) | 30 (53.8%) | 25 (67.6%) | 18 (52.9%) | |
| 50/0 | 5 (6.2%) | 20 (35.7%) | 10 (27.0%) | 7 (20.6%) | 0.341 |
| Lymph boost | | | | | |
| Yes | 14 (12.3%) | 27 (48.2%) | 19 (51.4%) | 14 (41.2%) | |
| No | 67 (82.7%) | 29 (51.8%) | 18 (48.6%) | 20 (58.8%) | 0.593 |

WP, whole pelvis; PAN, para-aortic lymph nodes; HDR-ICBT, high-dose-rate intracavitary brachytherapy; CS, center shield. Mann-Whitney U test was applied for comparison.

**Table 5. Multivariate Cox regression analysis for PIF.**

| Factors | HR | 95% CI | p-value |
|---|---|---|---|
| Postmenopausal state | 7.391 | 2.63–20.79 | <0.001 |
| BMI < 23 kg/m$^2$ | 0.673 | 0.741–1.572 | 0.156 |
| FIGO stage (III/IV vs. I/II) | 1.057 | 0.609–1.836 | 0.843 |
| Institution C (vs. A) | | | |
| Institution B | 0.916 | 0.864-1.120 | 0.897 |
| Institution C | 1.045 | 0.892-1.200 | 0.566 |
| Institution D | 1.214 | 1.015-1.412 | 0.035 |
| Institution (B/D vs. A/C) | 1.870 | 1.069–3.269 | 0.028 |

PIF, pelvic insufficiency fracture; HR, hazard ratio; BMI, body mass index

Our results are largely consistent with existing literature, which indicates that PIFs frequently occur as a late complication of pelvic RT in cervical cancer, particularly among postmenopausal women and individuals with decreased bone mineral density [1–9]. However, menopausal status should be interpreted cautiously, as it likely represents a surrogate marker for underlying bone health rather than a direct biological determinant of fracture risk. Baseline bone mineral density data were not available in this retrospective dataset, which represents an important limitation. Therefore, the strong association observed between menopausal status and PIF may partly reflect residual confounding related to undiagnosed osteoporosis or other skeletal fragility factors. The incidence observed in our cohort (28.4%) was within the upper range of the previously documented reports [1–5]. By implementing a standardized diagnostic protocol across participating institutions and restricting analysis to patients receiving comparable RT regimens, we attempted to reduce inter-study heterogeneity commonly observed in single-institutional analyses.

The timing of PIF onset exhibited a consistent pattern, with a median time to diagnosis of 12 months post-RT and 81.4% of cases occurring within the first two years. This observation aligns with current understanding that RT-induced bone fragility manifests within a relatively short latency period, potentially related to radiation-induced vascular injury, osteoblast dysfunction, and impaired bone remodeling [3,11]. However, our observation that some patients developed PIF beyond two years post-treatment suggests that prolonged surveillance may be warranted in selected high-risk populations.

A notable finding of this study was the significant inter-institutional variability in PIF incidence, even under largely comparable RT protocols and geographic proximity. The treating institution likely reflects complex unmeasured confounding factors rather than functioning as a direct causal determinant of fracture risk. Potential contributors include differences in imaging surveillance intensity, variations in CT or MRI acquisition protocols, differences in thresholds for reporting asymptomatic fractures, and subtle variations in treatment planning or delivery techniques that were not fully captured in protocol summaries. Therefore, institutional variation should be interpreted as a hypothesis-generating observation rather than definitive evidence of causality. Institution D, which demonstrated the highest PIF incidence, also had a higher proportion of patients receiving PAN irradiation.

Although fractures frequently occurred within irradiated regions, quantitative bone-specific dose–volume parameters were not available in this study. Consequently, while the spatial overlap between fracture sites and high-dose irradiation fields suggests a potential association, the absence of detailed dosimetric bone analysis precludes mechanistic interpretation of radiation dose effects on skeletal structures. Future studies incorporating standardized bone dosimetry will be necessary to clarify dose–response relationships.

Our findings carry substantial clinical implications. Given the high incidence of PIFs and their impact on survivors' quality of life, early identification of high-risk patients remains essential. These results should be interpreted as supporting

the need for individualized survivorship risk assessment rather than implying direct institutional causality. Baseline bone health assessment, including bone mineral density evaluation, may represent a potentially valuable strategy for identifying patients at elevated fracture risk, particularly among postmenopausal women. However, prospective validation is required before recommending routine preventive interventions.

Several limitations of this study should be acknowledged. First, although the retrospective multicenter design enabled evaluation across multiple institutions, it remains susceptible to potential selection bias and information bias, including differences in imaging surveillance practices. Second, important patient-related variables influencing bone fragility, including baseline bone mineral density, nutritional status, physical activity, smoking history, and socioeconomic factors, were not systematically captured. These variables may have contributed to residual confounding in the risk factor analysis. Third, death and disease progression were treated as censoring events in time-to-event analyses. Because these outcomes may function as competing risks for PIF occurrence, alternative statistical approaches such as competing-risk regression models could provide more precise risk estimation. This represents an important methodological consideration for future investigations. Fourth, most patients in this cohort were treated using conventional four-field box techniques with central shielding, with relatively limited use of contemporary IMRT or VMAT approaches. As modern image-guided radiotherapy techniques are increasingly used to optimize dose distribution and reduce treatment-related toxicity, the generalizability of our findings to current clinical practice may be limited.

In conclusion, this multicenter study demonstrates that PIFs represent a frequent late complication following definitive RT for cervical cancer, with substantial inter-institutional variability in incidence despite broadly standardized treatment protocols. Postmenopausal status and treating institution were identified as associated factors; however, these findings should be interpreted cautiously given the absence of baseline bone health data and detailed dosimetric analysis. Our results highlight the need for further prospective studies incorporating bone density assessment and radiation dose–volume evaluation to better define risk stratification and preventive strategies for PIFs in cervical cancer survivors.

## Supporting information

**S1 Data. Minimal dataset of this study.**
(DOCX)

## Author contributions

**Data curation:** Noriaki Imafuku, Dan Yamamoto, Masae Yorimitsu, Junichi Kodama, Hisako Nagasaka, Yoshie Nakanishi.

**Formal analysis:** Shinsuke Shirakawa.

**Supervision:** Shoji Nagao, Kotaro Yoshio, Toshiharu Mitsuhashi, Hisashi Masuyama.

**Writing – original draft:** Shinsuke Shirakawa.

**Writing – review & editing:** Shinsuke Shirakawa.

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
