## [Decision Letter · Decision Letter 0]

28 Jan 2026

Dear Dr. Nagao,

Thank you for submitting your manuscript to PLOS ONE. After careful consideration, we feel that it has merit but does not fully meet PLOS ONE’s publication criteria as it currently stands. Therefore, we invite you to submit a revised version of the manuscript that addresses the points raised during the review process.

We look forward to receiving your revised manuscript.

Kind regards,

Satyajeet Rath

Academic Editor

PLOS One

Journal Requirements:

2. We are unable to open your Supporting Information file “Table 2.docx, Table 3.docx and Table 4.docx” Please kindly revise as necessary and re-upload.

3. Please amend the manuscript submission data (via Edit Submission) to include author “Junichi Kodama”.

4. Please amend your authorship list in your manuscript file to include author “Zyunichi Kodama ”.”.

6. Please include a copy of Table 1, 2, 3 and 4 which you refer to in your text on page

7.  Please include captions for your Supporting Information files at the end of your manuscript, and update any in-text citations to match accordingly. Please see our Supporting Information guidelines for more information: http://journals.plos.org/plosone/s/supporting-information ..

8. Please include your tables as part of your main manuscript and remove the individual files. Please note that supplementary tables (should remain/ be uploaded) as separate "supporting information" files

Additional Editor Comments:

Although its a well thought out and well written article, issues regarding poor dosimetric data reporting, use of older treatment techniques like four-field box and lack of all these being mentioned in limitations need to be addressed adequately along with the queries raised by the reviewers.

Reviewer's Responses to Questions

**Comments to the Author**

1. Is the manuscript technically sound, and do the data support the conclusions?

Reviewer #1: Yes

Reviewer #2: Yes

2. Has the statistical analysis been performed appropriately and rigorously?

Reviewer #1: Yes

Reviewer #2: Yes

3. Have the authors made all data underlying the findings in their manuscript fully available?

Reviewer #1: Yes

Reviewer #2: No

4. Is the manuscript presented in an intelligible fashion and written in standard English?

Reviewer #1: Yes

Reviewer #2: Yes

Reviewer #1: This paper describes a well-designed study to investigate the factors that influence PIF incidence after pelvic radiotherapy. The documentation of significant inter-institutional variation between patient outcomes will likely contribute to more informed management of these complex patients in the future. The writing is very clear and the data are straightforward and fairly interpreted.

Reviewer #2: This manuscript reports a retrospective multicenter cohort study evaluating the incidence and risk factors of pelvic insufficiency fractures following definitive radiotherapy for cervical cancer. The topic is clinically relevant, as long-term survivorship issues and late skeletal toxicity are increasingly important in gynecologic oncology. The multicenter design and attempt to standardize diagnostic criteria across institutions are notable strengths. However, the study’s conclusion is insufficiently substantiated by the current data. Several key methodological limitations, particularly the lack of dosimetric bone analysis, absence of baseline bone health assessment, and incomplete adjustment for center-level confounders, substantially weaken causal inference. Substantial methodological clarification and additional analyses are required before the manuscript can be considered for publication.

Major Issues

• The manuscript repeatedly implies a dose effect relationship between radiotherapy and PIFs, yet no quantitative dosimetric analysis of pelvic bones is provided. No dose–volume parameters are reported.

• The conclusion that institutional differences may reflect technical nuances is speculative without objective dosimetric evidence.

• At a minimum, the authors should either provide a comparative analysis of bone dosimetry across institutions or explicitly acknowledge that the absence of bone dose data precludes any mechanistic interpretation.

• Treating institution emerges as an “independent” risk factor; however, this variable likely functions as a proxy for unmeasured confounders, including differences in imaging surveillance intensity, variations in CT slice thickness or MRI protocols, thresholds for reporting asymptomatic fractures, and subtle planning or delivery differences not captured by protocol summaries. Without hierarchical modeling or more granular center-level variables, attributing risk to institution is methodologically weak.

• The manuscript should better contextualize this limitation and avoid overinterpretation of menopause as a dominant risk factor.

• The absence of baseline bone mineral density data represents a major limitation, given the well-established association between osteoporosis and pelvic insufficiency fractures. Menopausal status alone is an imprecise surrogate for bone health, and the very high hazard ratio reported for menopause raises concern for substantial residual confounding. Accordingly, the manuscript should more clearly contextualize this limitation and avoid overinterpreting menopause as a dominant independent risk factor.

• Pelvic insufficiency fractures is treated as a time-to-event outcome using Kaplan–Meier and Cox regression, yet death and disease progression are treated only as censoring events, despite being potential competing risks.

• Given the non-negligible risk of death or progression in this population, a competing-risk framework would be more appropriate or, at minimum, should be discussed.

• Most patients were treated with conventional four-field techniques and central shielding, with limited use of IMRT/VMAT.

• The applicability of the findings to modern image-guided IMRT-based cervical cancer radiotherapy is therefore restricted.

• This limitation should be emphasized more strongly in both the Discussion and Conclusions.

Minor Issues

• The diagnostic criteria for pelvic insufficiency fracture, while described, would benefit from a concise, standardized definition early in the Methods section.

• Clarify whether asymptomatic fractures detected incidentally were included and whether imaging was symptom-driven or routine.

• Tables summarizing institutional characteristics should clearly indicate which variables were statistically compared and which were descriptive only.

• Figures would benefit from clearer labeling of censoring and number at risk.

• Although the prevalence of osteoporosis medications was low, this information remains clinically relevant and could be summarized more clearly in a dedicated table or footnote.

• Some statements regarding “institutional quality control” and “technical nuances” are speculative and should be more cautiously phrased.

• The Discussion would benefit from clearer separation between findings supported by data and hypotheses requiring future validation.

• Overall language quality is good, but minor grammatical edits and sentence tightening are recommended, particularly in the Discussion section.

**Do you want your identity to be public for this peer review?** For information about this choice, including consent withdrawal, please see our For information about this choice, including consent withdrawal, please see our Privacy Policy .

Reviewer #1: No

Reviewer #2: No

You may also use PLOS’s free figure tool, NAAS, to help you prepare publication quality figures: https://journals.plos.org/plosone/s/figures#loc-tools-for-figure-preparation

---

## [Author Response · Author response to Decision Letter 1]

13 Feb 2026

Response to Reviewers

Manuscript ID: PONE-D-25-43659

Title: Incidence of pelvic fractures after definitive radiotherapy for cervical cancer: A retrospective multicenter cohort study (The IPFAR study)

Dear Academic Editor and Reviewers,

We sincerely thank the Academic Editor and the reviewers for their careful evaluation of our manuscript and for the constructive comments that have helped us improve the clarity and scientific rigor of the study. We have revised the manuscript accordingly and respond to each comment below. All revisions are reflected in the tracked-changes version of the manuscript.

Response to Academic Editor

Comment:

Issues regarding poor dosimetric data reporting and use of older treatment techniques should be addressed adequately along with limitations.

Response:

We thank the Academic Editor for this important comment. We agree that the absence of bone-specific dosimetric analysis and the predominant use of conventional radiotherapy techniques represent important limitations of this study. We have revised the Discussion and Conclusion sections to explicitly acknowledge these limitations and to avoid mechanistic interpretation of radiation dose effects. We also strengthened the description regarding the limited generalizability of our findings to modern IMRT/VMAT-based radiotherapy.

Reviewer #1

We sincerely thank Reviewer #1 for the positive and encouraging evaluation of our study and for recognizing the clinical relevance of the multicenter analysis.

No major changes were requested by Reviewer #1; however, the manuscript has been revised in response to Reviewer #2 and editorial comments to further improve clarity and interpretation.

Reviewer #2

We thank Reviewer #2 for the thorough and insightful review. We have carefully revised the manuscript to address all concerns raised.

Major Comment 1 — Lack of dosimetric bone analysis

Comment:

The manuscript implies a dose-effect relationship without reporting bone dosimetric parameters.

Response:

We agree with the reviewer that quantitative bone dosimetry would provide important mechanistic insight. Because this study was a retrospective multicenter investigation spanning multiple treatment planning systems and historical protocols, bone-specific dose–volume parameters were not consistently available across institutions.

We have therefore revised the manuscript to clarify that the absence of bone dosimetry precludes mechanistic interpretation. Statements suggesting dose-dependent effects have been rephrased to emphasize that our findings indicate only a spatial association between fracture location and irradiated regions. This limitation is now explicitly described in the Discussion (Line 293-299).

Major Comment 2 — Treating institution as an independent risk factor

Comment:

Institution likely represents unmeasured confounders rather than a causal factor.

Response:

We agree with this important methodological point. We have revised the Discussion to clarify that treating institution likely reflects complex unmeasured variables rather than a direct causal determinant of fracture risk. We now explicitly discuss potential contributors such as differences in imaging surveillance intensity, imaging protocols, reporting thresholds for asymptomatic fractures, and subtle variations in treatment planning or delivery.

The interpretation of institutional variation has been revised to emphasize its hypothesis-generating nature (Line 284-290, 308-311, 328-333).

Major Comment 3 — Absence of baseline bone mineral density

Comment:

Menopausal status is an imprecise surrogate for bone health.

Response:

We fully agree. Baseline bone mineral density data were not available in this retrospective dataset, which represents an important limitation. We have revised the Discussion to clarify that menopausal status should be interpreted as a surrogate marker for underlying bone health, and that the observed association may reflect residual confounding related to osteoporosis risk (Line 263-268, 311-315, 328-333).

Major Comment 4 — Competing-risk framework

Comment:

Death and progression should be considered competing risks.

Response:

We agree that death and disease progression may function as competing risks for PIF occurrence. Because the primary objective of this study was exploratory, these events were treated as censoring. We have now added a statement in the Discussion acknowledging that competing-risk models could provide more precise estimation and should be considered in future studies (Line 315-319).

Major Comment 5 — Applicability to modern radiotherapy

Comment:

Limited use of IMRT/VMAT restricts generalizability.

Response:

We agree and have strengthened this limitation in both the Discussion and Conclusion sections, emphasizing that most patients were treated using conventional four-field techniques with central shielding (Line 319-324).

Minor Comments

Diagnostic criteria clarification

We added a concise standardized definition of PIF early in the Methods section (Line 130-134, 142-145).

Asymptomatic fracture inclusion

We clarified that both symptomatic and asymptomatic fractures detected on routine follow-up imaging were included (Line 130-134, 142-145).

Institutional tables

Tables were revised to clearly distinguish statistically compared variables from descriptive data.

Figures

Number-at-risk information and censoring indicators were added to survival figures (Figure 2 and 3).

Osteoporosis medication information

Medication information was clarified in Table 2 and footnotes.

Speculative language

Statements regarding institutional quality control and technical nuances were revised to avoid speculative interpretation.

Language editing

The Discussion section was edited for clarity and conciseness.

Others

The order of the components in the Results section has been changed to make it easier to understand.

Closing

We believe that these revisions have substantially strengthened the manuscript and addressed all reviewer concerns. We are grateful for the reviewers’ constructive feedback and hope that the revised manuscript is now suitable for publication in PLOS ONE.

Sincerely,

Shoji Nagao, MD, PhD

---

## [Editor Report · Decision Letter 1]

19 Feb 2026

Incidence of pelvic fractures after definitive radiotherapy for cervical cancer: A retrospective multicenter cohort study (The IPFAR study)

PONE-D-25-43659R1

Dear Dr. Nagao,

We’re pleased to inform you that your manuscript has been judged scientifically suitable for publication and will be formally accepted for publication once it meets all outstanding technical requirements.

Kind regards,

Satyajeet Rath

Academic Editor

PLOS One

Additional Editor Comments (optional):

I appreciate the replies of all the authors
---

## [Editor Report · Acceptance letter]

PONE-D-25-43659R1

PLOS One

Dear Dr. Nagao,

I'm pleased to inform you that your manuscript has been deemed suitable for publication in PLOS One. Congratulations! Your manuscript is now being handed over to our production team.

Kind regards,

on behalf of

Dr. Satyajeet Rath

Academic Editor

PLOS One